# New Unexpected Species of *Acheta* (Orthoptera, Gryllidae) from the Italian Volcanic Island of Pantelleria

Bruno Massa [1,*], Camillo Antonino Cusimano [2], Paolo Fontana [3,4] and Cesare Brizio [4]

1 Department of Agriculture, Food and Forest Sciences, University of Palermo, Viale delle Scienze, 13-90128 Palermo, Italy
2 Stazione Ornitologica Aegithalos, Via Aquino Molara, 75-90046 Palermo, Italy
3 Fondazione Edmund Mach di San Michele all'Adige—Centro Trasferimento Tecnologico, Via E. Mach, 1-39098 Trento, Italy
4 World Biodiversity Association, Museo Civico di Storia Naturale di Verona, Lungadige Porta Vittoria, 9-37129 Verona, Italy
* Correspondence: bruno.massa@unipa.it

**Abstract:** In late April 2022, while listening to audio files from an unsupervised bioacoustic assessment of the shearwater populations (Aves, Procellariiformes) on the coast of Pantelleria island (Sicily, Italy), a cricket song of unknown attribution was heard. The first bioacoustic analyses, including FFT-based spectrograms and sound pressure envelopes, confirmed that it could not be attributed to the known sound of any Italian nor Mediterranean species of cricket. In the ensuing weeks, field research at the original station and further localities on the southern coast of Pantelleria provided photographs, living specimens, and further audio records. As soon as the photos were shared among the authors, it became clear the species belonged to the genus *Acheta*. Further bioacoustic analyses and morphological comparison with type specimens of Mediterranean and North-African congenerics in relevant collections and the scientific literature were conducted: they confirmed that the findings could only be attributed to a still undescribed species that escaped detection due to its impervious and unfrequented habitat. *Acheta pantescus* n. sp. is apparently restricted to the effusive coastal cliffs of the island of Pantelleria, a habitat whose scant extension and vulnerability require environmental protection actions such as the inclusion in a special Red List by the IUCN Italian Committee.

**Keywords:** new species; biogeography; Mediterranean; red list; bioacoustics

## 1. Introduction

On the night of 27 April 2022, during a survey aimed at the assessment of the presence of the pelagic birds Scopoli's shearwater *Calonectris diomedea* (Scopoli, 1769) and Yelkouan shearwater *Puffinus yelkouan* Acerbi, 1827 (Aves, Procellariiformes) on the island of Pantelleria (Sicily, Italy), BM and CC placed a Wildlife Acoustics sound recorder, set in unsupervised mode from sunset to dawn, near the cliffs of Punta Limarsi on the south-eastern coast of the island. After retrieving the recorder, they examined the content of the SD card. They noticed an Orthopteran song with Gryllinae affinities that, to the unaided ear, seemed not to match any species reported from Pantelleria or other Italian areas.

The recording was shared the same day with the other authors. Between 27 April and mid-May, with only acoustic evidence available, pursuant to the principle of parsimony, the authors engaged in a series of attempts to attribute the recorded song to a known species of cricket. With many decades of field experience in Orthopteran research, to ascertain the lack of comparable audio samples of Italian cricket songs, BM and PF could both rely on their first-hand knowledge and the systematical comparative hearing of the new song and that of the Italian Gryllidae, as available respectively on DVD-ROM [1] and CD-ROM [2] on the most exhaustive Orthopteran sound collections for the Italian territory that they co-authored. Independently, CB performed similar comparative investigations, including

his own published and unpublished audio material. Some similarities with the song of *Gryllus bimaculatus* De Geer 1773, present in Pantelleria, were initially observed and quickly excluded among the known species in the Mediterranean area for which audio samples were obtained from online sources; also, the "silver-bell cricket" *Gryllodinus kerkennensis* (Finot, 1893) was considered for a while as a plausible candidate.

After the serendipitous recording, during a subsequent expedition to Pantelleria island, on 14–15 May, BM, CC, and PF carried out further audio takes. They were respectively obtained at the cliffs of Punta Limarsi lighthouse and two kilometers northeast, at the cliff of Martingana. During the same visit, they captured three living specimens (a female nymph and an adult male near the lighthouse of Punta Limarsi and an adult female at Martingana). The elytron venations visible in the photographs showed a striking similarity with those of the known species of the genus *Acheta* Fabricius, 1775, as illustrated by Gorochov [3].

With the bioacoustic investigation underway, the emergence of pictures and specimens in mid-May re-oriented the attention towards that genus. While the morphological investigations progressed, acoustic comparisons were then restricted to the species of *Acheta*, whose bioacoustical coverage—as better elucidated in the section describing the affinities of the novel species' song—is very scanty, both in literature and in online audio repositories. As soon as the case for a new species became undisputable on morphological bases, bioacoustical analyses were exclusively aimed at an exhaustive description of the sound emissions of the new cricket from Pantelleria.

On 5–6 July 2022, further research by BM and CC resulted in higher quality recordings but couldn't provide any further specimens, as the insects sang from deep crevices on the cliff faces. The presence of the new species seemed more widespread than previously estimated but ecologically constrained to the cliffs, with no observation nor recording obtained in other environments.

## 2. Materials and Methods

### 2.1. Discovery of the Cricket

As reported in the Introduction, this new species of *Acheta* was discovered by chance while listening to the recording obtained from a Wildlife Acoustics sound recorder placed on the night of 27 April 2022 near the lighthouse of Punta Limarsi of Pantelleria (See Table 1 for a list of relevant locations).

**Table 1.** List of localities cited in the text, with dates and kind of evidence collected.

| Map Ref. No. | Location | Dates of Visit | Audio Recording | Specimen Collected |
|---|---|---|---|---|
| 1 | Cimitero Scauri | 5–6 July 2022 | ☑ | ☐ |
| 2 | Salto la Vecchia | 5–6 July 2022 | ☑ | ☐ |
| 3 | Balata dei Turchi | 27 April 2022–14–15 May 2022 | ☑ | ☐ |
| 4 | Punta Limarsi lighthouse | 27 April 2022–14–15 May 2022–5–6 July 2022 | ☑ | ☑ |
| 5 | Punta Limarsi | 27 April 2022–14–15 May 2022–5–6 July 2022 | ☑ | ☐ |
| 6 | Cala Rotonda—Martingana | 14–15 May 2022 | ☑ | ☑ |
| 7 | Punta del Formaggio | 5–6 July 2022 | ☑ | ☐ |

Later, during the nights of 14–15 May, it was possible to discover a second site, in the locality Martingana, where many cricket individuals were singing. On the same days, one male and one female adult were collected at the Punta Limarsi lighthouse and Martingana, respectively. Finally, during the night of 5–6 July 2022, Wildlife Acoustics sound recorders allowed to discover the presence of the same cricket in the localities Cimitero di Scauri, Salto la Vecchia, Punta Limarsi, and Punta del Formaggio. On the same night at the lighthouse of Punta Limarsi and Martingana, it was possible to observe the habitat of the crickets, which were singing from the crevices of volcanic rocks just a few meters above sea level, and in one case from a drystone wall behind a big *Juniperus turbinata* (Gussone) tree. Due to the

shyness of the cricket, it was not possible to collect other specimens, but we successfully recorded its song with an Edirol R09 recorder.

### 2.2. Morphological Analysis

We examined specimens of different species of the genus *Acheta* preserved in the museums of Genoa and Madrid and the private collection of the first author. Collected specimens and their parts were photographed with a Nikon Coolpix 4500 digital camera, mounted on a Wild M3 Stereomicroscope, and with an Olympus Stylus TG-5 Tough (cf. [4]). Photographs were integrated using the freeware CombineZP [5]. Mounted specimens were measured with a digital caliper (precision 0.01 mm). The following measurements were taken (in mm): Body length—dorsal length from the head to the apex of the abdomen; Pronotum length; Tegmina length; Hind femora length; Ovipositor—maximum length, subgenital plate included.

### 2.3. Abbreviations for Collections and Museums

BMPC: Bruno Massa personal collection, Palermo, Italy.
MNCN: Museo Nacional Ciencias Naturales, Madrid, Spain.
MSNG: Museo Civico di Storia Naturale 'G. Doria', Genoa, Italy.
MSNR: Museo Civico di Storia Naturale, Rovereto (Trento), Italy.

### 2.4. Recording and Audio Analysis Equipment

Table 1 summarizes the audio samples considered in this study.

Even though the temporal features of the recordings obtained with different devices and settings may be comparable regardless of the equipment, issues including differences in microphones among the recorders, use of the lossy format by the smartphone, and differences in bandwidth and sampling frequencies may result in an inconsistent representation of the frequency domain by the different devices.

Considering that some types of songs observed (spring calling/advertising; summer calling/advertising; "type 2") were recorded by just one kind of device, bioacoustic analyses include all the audio material available, regardless of the recording technique, to ensure a description as exhaustive as possible of the different songs observed from late April to the first week of July.

Baker & Chesmore [6] standardized the bioacoustic terminology for insects. Their well-placed effort acknowledges the existence of a few contradictions in terminology, e.g., about the concepts of "pulse", "pulse train", and "bout". For the sake of clarity and simplicity, we will strive to adhere to the terminology they propose and, in song descriptions, we will restrain to the following terms:

**Pulse**—indivisible unit of sound, typically corresponding to a single tooth impact.

**Syllable**—single complete stridulatory movement (the opening and closing of the elytra in Ensifera, the up and down motion of the femora against the elytra in some Acrididae).

**Echeme**—first-order assemblage of syllables.

**Echeme-Sequence**—first-order assemblage of echemes (may include individual syllables that precede or follow the echeme).

Furthermore, the following unambiguous terms, also cited by Baker & Chesmore [6] as conveying meaning (particularly for human identification by ear) but not recognized as part of their controlled vocabulary), will be used:

**Trill**—a more or less long echeme made by subequal, closely spaced syllables.

Description will also require the use of terminology from Buzzetti & Barrientos-Lozano [7], in particular, related to the concept of "pulse train":

**Pulse train (PT):** First order grouping of more than one pulse, preceded and followed by a silent or nearly so interval lasting longer than any intervals between the pulses.

**Pulse train group (PTG):** Second order grouping of pulses comprising two or more PTs produced in succession.

**Major pulse train (MaPT):** A pulse train of longer and usually greater intensity. In Ensifera, it is generally produced on wing closure.

**Minor pulse train (MiPT):** A pulse train of shorter and usually lesser intensity. In Ensifera, it is generally produced on wing opening.

The April recordings (PCM WAV files) were obtained by a Wildlife Acoustics "Song Meter Micro" digital weatherproof recorder with a built-in microphone set at its default sampling frequency of 22,050 Hz. The device was optimized for long duration, medium quality soundscape recordings, for supervised playback aimed at detecting bird songs and could not provide audio files suitable for in-depth bioacoustic analyses; nevertheless, it allowed to exclude the attribution of the song to any known Italian Gryllidae.

On 14–15 May, in Martingana, also a Samsung SM-A750SN Smartphone was used as an audio recorder via its built-in application. It provided recordings (lossy, compressed m4a format) hard-limited to around 16 kHz that were converted to mp3 for the acoustic analyses. Higher quality recordings (PCM WAV files) were obtained on 5–6 July 2022, using an Edirol R09 digital recorder with built-in stereo microphones at a sampling frequency of 44.1 kHz.

Sound description includes pressure envelopes (relative pressure in dB full scale), time/frequency spectrograms, and frequency/sound pressure analysis diagrams, generated on an ASUS EeePC 1225B Netbook (for the July recordings, on a Gigabyte Brix desktop computer) by Adobe Audition 1.0 software running under Windows 10 64bit operating system. The same software was used to apply a 1700 Hz high-pass software filter to exclude the background noise (wind, sea waves, anthropogenic sounds) from the April and May recordings. This intervention preserved all the sound components attributable to the new species.

The frequency resolution of FFT-based analyses is directly proportional to FFT size, see [8,9]). Concerning the expected size of the pictures in this article, an FFT size of 8192 byte was chosen as the best compromise between detail and smoothness of the picture. Frequency analyses were generated with the Blackman-Harris method [10–12] by scanning a continuous interval of the audio samples as per Table 2.

**Table 2.** Details about the audio samples used for the analyses.

| DEVICE | Record Date | Sampling Frequency | Locality | Duration | Song Type | Interval Analysed for Frequency/Sound Pressure |
|---|---|---|---|---|---|---|
| Song Meter Micro | 27 April 2022 | 22.05 kHz | Punta Limarsi | 30′00″.000 | Calling | 2 s |
| Song Meter Micro | 27 April 2022 | 22.05 kHz | Punta Limarsi | 1′52″.676 | Calling | 2 s |
| Song Meter Micro | 27 April 2022 | 22.05 kHz | Punta Limarsi | 2′53″.493 | Calling | 2 s |
| Samsung SM-A750SN | 15 May 2022 | 32.00 kHz | Martingana | 1′30″.958 | Type 2 | 2 s |
| Edirol R09 | 6 July 2022 | 44.1 kHz | Martingana | 1′46″.701 | Calling | 10 s |

Bioacoustic illustrations are based on Adobe Audition screenshots, post-produced with Adobe Photoshop Elements, as needed to ensure optimal contrast. Contrast enhancement was also applied to photographs of the tegmina and genitalia to generate line drawing-like images. MS-Paint was used to draw custom horizontal/vertical reference rulers. Visual improvements did not alter the data or the analysis results.

Quantitative analyses were performed in Adobe Audition. Echeme repetition rates and pulse counts were obtained manually from the pressure envelopes of the required duration, directly on screen or from printed screenshots.

The entire frequency range available in each recording was thoroughly analyzed, but to optimize informativeness, the frequency/pressure observation window was customized in each picture: sound pressure axes are limited at the top by the highest volume observed and at the bottom by the bottom noise at any frequency. Frequency axes are limited at

the top by the actual useful frequency range and at the bottom frequency by the lowest components attributable to the singing specimen.

Even though the team wasn't equipped with a portable thermometer, meteorological data about Pantelleria on the visit dates were subsequently collected from the local meteorological station and are provided in Table 3.

**Table 3.** Meteorological data at the time of visit in the localities listed in Table 1, as collected by the Pantelleria meteorological station at Madonna delle Grazie (Longitude: 11.953109, Latitude: 36.793378).

| Date | Collection Time | Hourly Average Air Temperature (°C) | Hourly Average Relative Humidity (%) | Instantaneous Wind Speed at 2 m (m/s) |
|---|---|---|---|---|
| 27 April 2022 | 21:00 | 18 | 37 | 1 |
| 14 May 2022 | 21:00 | 16.1 | 57 | 0.9 |
| 15 May 2022 | 21:00 | 23.1 | 25 | 1 |
| 6 July 2022 | 21:00 | 25.5 | 81 | 3.4 |

*2.5. Crowdsourcing of the New Species Name*

As soon as it became evident that the population of *Acheta* living on the coasts of the island belonged to an undescribed species, an opinion poll was launched by the Parco Nazionale di Pantelleria both on the website of the Park and on Instagram, as well as on the Forum entomologiitaliani.net, offering the opportunity to choose the name of the newly discovered species from among four alternatives:

1. *pantescus*, adjective indicating the inhabitants of Pantelleria;
2. *marinus*, indicating the peculiar eco-ethology of the species, linked to the rocks on the sea level;
3. *petrosus*, as an alternative to the previous one;
4. *phantasma*, as a noun in apposition, indicating the difficulty of observing this cricket, whose song was recorded only by chance.

**3. Results**

*3.1. Acheta pantescus n. sp., English Name: Pantelleria Cricket*

lsid:*zoobank.org:act:D7FDFE7F-D8D2-411E-B795-2827B378DEA7*.

Species Name

In order of preference, the total 700 ballots, almost entirely collected on the website and Instagram pages of the Parco Nazionale di Pantelleria, were divided as follows:

- *pantescus*, 396 votes (56.6%);
- *phantasma*, 146 votes (20.8%);
- *petrosus*, 136 votes (19.4%);
- *marinus*, 22 votes (3.1%).

*3.2. Song Description*

3.2.1. Calling/Advertising Song

The song is emitted at night, at a rate of up to 10 syllables/s at 18 °C. Syllable duration ranges between 30 and 50 ms. Echemes vary in duration and number of syllables, and tend to assume the shape of uniform trills with a duration of up to 20 or more seconds. Prolonged trills may begin in a slight crescendo involving 1–5 syllables and end abruptly. Besides the trills, echeme-sequences may include one or two feebler final syllables. They may also unorderly include shorter groups of syllables, the first two–four in crescendo (increasing volume). Sound pressure envelope illustrations include diverging red lines above and under the group of syllables in crescendo.

Figure 1 covers the recordings obtained in the open field on 27 April—reported air temperature on Pantelleria was around 18 °C (see Table 3): in sound pressure envelopes (Figure 1A–C), each syllable in a trill includes a very short final pulse, lasting around 5 ms, with a volume around 60% lower than the syllable's maximum. Pulse count is around 150 for the major pulse train, around 30 for the minor pulse train, and sometimes a feeble tail consisting of 10–15 pulses is observed.

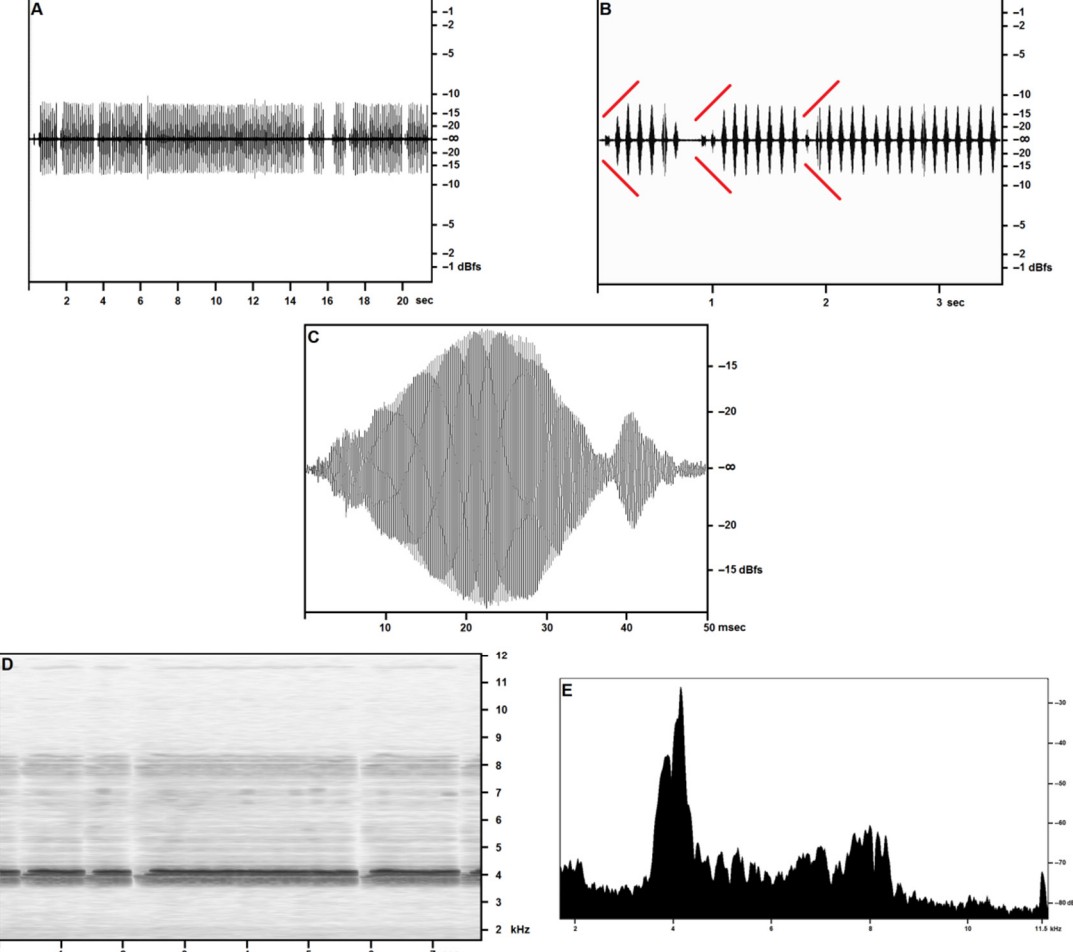

**Figure 1.** *Acheta pantescus* n. sp., calling/advertising Song—(**A**), Sound Pressure Envelope, 21.5 s; (**B**), Sound Pressure Envelope, 3.5 s—diverging red lines highlight crescendos; (**C**), Pulses (tooth strikes), one syllable; (**D**), Time/Frequency Spectrogram, 7.7 s; (**E**), Frequency/Sound Pressure analysis, scan on two seconds.

The opening and closing syllables in each trill may show a different structure, consisting of up to three distinct pulse trains of subequal duration, the middle pulse train loudest, or two pulse trains, respectively longer and louder, and shorter and feebler. Trills may occasionally include lower volume and structured syllables, as described above.

At an air temperature of 18 °C, a time-frequency spectrogram (Figure 1D) shows a well-defined fundamental band with peak pressure concentrated at around 4100 Hz with a secondary, wider peak centered at around 4000 Hz and a slightly lower pressure band between around 3750 Hz and 3900 Hz. The first harmonic band reproduces the same structure more feebly and less distinctly, at double the frequencies reported for the fundamental. A narrow band may be observed near the cut-off frequency, and its disappearance in the wider band recordings suggests considering it a technogenic artifact.

The frequency/acoustic pressure analysis (Figure 1E) based on the full scan of 2 s (around 20 syllables) of calling a song by a single individual confirms the presence of the

three-peaked band as per the time/frequency spectrogram. The same analysis performed on single syllables (not illustrated here) provides a single-peaked fundamental, demonstrating a slight inconsistency in the fundamental frequency in adjacent syllables emitted by the same individual.

The calling/advertising song was recorded in higher quality in July (Figure 2) at an air temperature of 25.5 °C from specimens singing from crevices in the cliff. Song structure was smoother (higher overall similarity among syllables) (Figure 2A–C), higher-pitched (Figure 2D) (consistently with a higher temperature than in April recordings), and didn't include the structured syllables nor the final pulse train observed in the spring songs. Frequency/acoustic pressure analysis (Figure 2E) based on a 10-s scan of three subsequent trills, confirms the presence of a two-peaked fundamental band, with a primary narrow spike at around 5154 Hz (−23.51 dBfs) and a secondary spike at 5028 Hz (−28.46 dBfs). Primary and secondary fundamental repeat at regular harmonic intervals, respectively, in very narrow spikes, with exceptional consistency (less than 0.4% frequency variance if compared with the natural arithmetical multiples of the fundamental) and in progressively blunter humps (the highest variance is shown by the fourth harmonic, and the third secondary harmonic equal in volume to the third primary harmonic). Numerical data based on the sample size of 10 s are provided in Table 4.

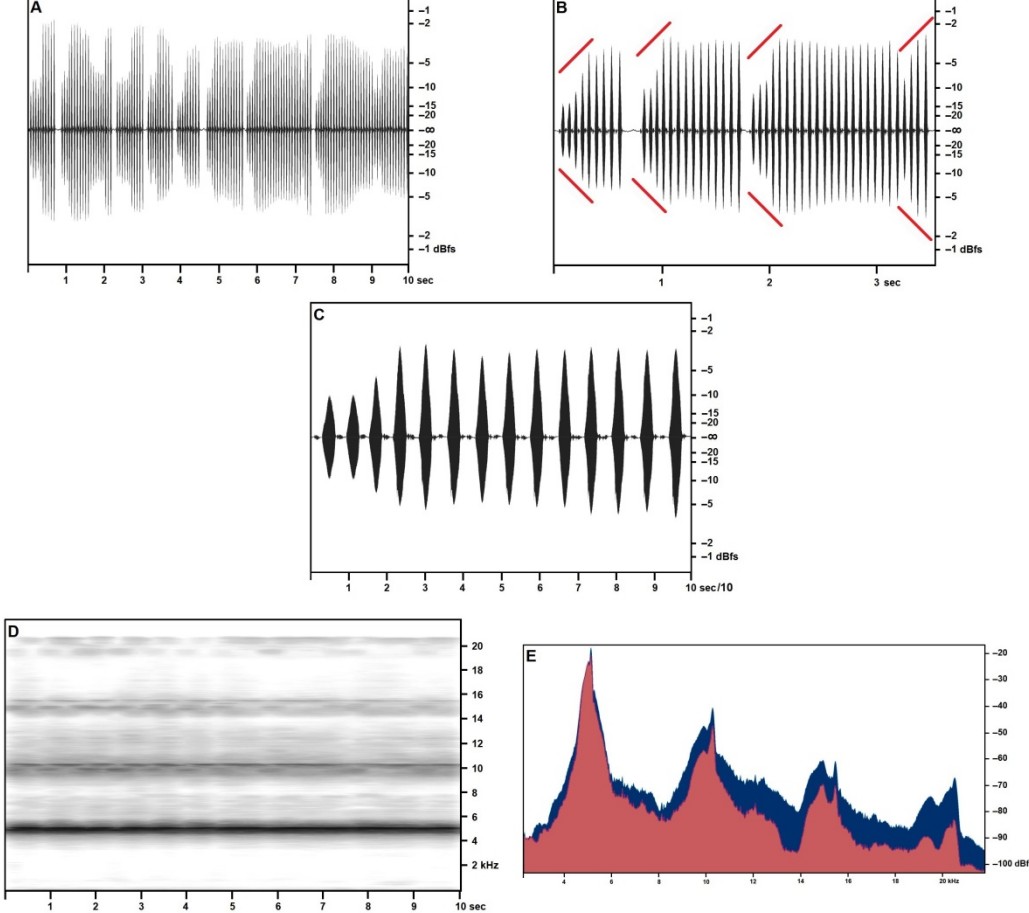

**Figure 2.** *Acheta pantescus* n. sp., calling/advertising Song, 44.1 kHz sampling frequency stereo recording—(**A**), Right Channel—Sound Pressure Envelope (10 s); (**B**), Right Channel—Sound Pressure Envelope (3 s)—diverging red lines highlight crescendo; (**C**), Right Channel—Sound Pressure Envelope, 1-s trill; (**D**), Time/Frequency Spectrogram, 10 s; (**E**), Frequency/Sound Pressure analysis, scan on ten seconds, Right Channel in the foreground (red).

**Table 4.** Harmonic structure of the July calling/advertising song of *Acheta pantescus* n. sp. in the high-quality stereo recordings, based on a sample size of 10 s.

| Element | | Frequency (Hz) | Theoretical Value | Δ | Pressure (dBfs) |
|---|---|---|---|---|---|
| Fundamental | Harmonic | | | | |
| Primary | I | 5154 | – | – | −23.51 |
| | II | 10,270 | 10,308 | +0.37% | −45.37 |
| | III | 15,440 | 15,462 | +0.14% | −65.44 |
| | IV | 20,540 | 20,616 | +0.37% | −70.39 |
| Secondary | I | 5028 | – | – | −28.46 |
| | II | 9894 | 10,056 | +1.61% | −51.8 |
| | III | 14,960 | 15,084 | +0.83% | −64.02 |
| | IV | 19,500 | 20,118 | +3.14% | −76.92 |

### 3.2.2. "Type 2" Song

The "type 2" song, reported from the cliff face of Martingana on the night of 15 May, with an air temperature of 23.1 °C, is illustrated in Figure 3 and shows short echemes of 6–12 syllables (Figure 3A,B) arranged in a crescendo up to 60% of the maximum volume, followed by two or three subequal syllables at maximum volume. Echemes are emitted in close sequence, and two or more echemes may merge, still preserving the cycle of crescendo and maximum volume syllables.

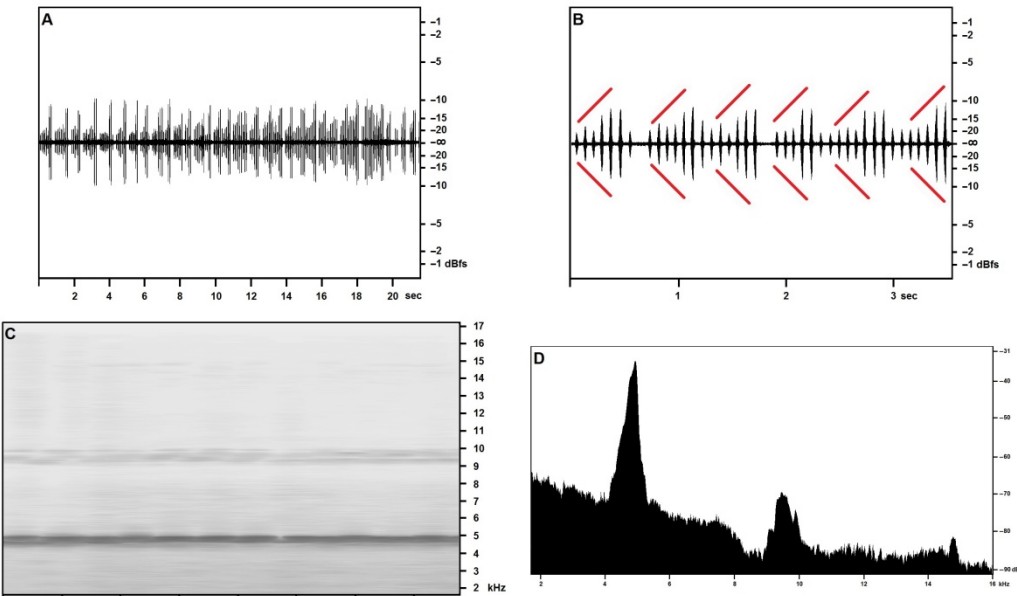

**Figure 3.** *Acheta pantescus* n. sp., "type 2" Song—(**A**), Sound Pressure Envelope, 21.5 s; (**B**), Sound Pressure Envelope, 3.5 s—diverging red lines highlight crescendo; (**C**), Time/Frequency Spectrogram; (**D**), Frequency/Sound Pressure analysis, scan on two seconds.

The song, although surely attributable to the same species, shows an entirely different pattern than the calling/advertising song, a difference that—pending further investigation—may be reasonably attributed to a different function, such as courtship, a hypothesis that may be supported by the disappearance of this type of song in the July recordings from the same locations.

Both the time-frequency spectrogram of around 8 s of the song (Figure 3C) and the frequency analysis of two seconds of the same song (Figure 3D) confirm a fundamental frequency of around 4800–4900 Hz, much nearer to the fundamental observed in the July calling/advertising song: this frequency peak slopes gently towards the lower frequencies, and more abruptly towards the higher range.

The first harmonic band is well recognizable and blunter, with a clear spike at around 10 kHz. The narrow second harmonic band appears at around 14.9 kHz, in good accord with the fundamental.

### 3.2.3. Generalities and Bioacoustic Comparison with Other Acheta Songs

The song is decidedly High-Q [13,14]. The higher-pitched songs ("Type 2" recorded on 15 May and July calling/advertising) were recorded at higher air temperatures (23.1° and 25 °C respectively, compared with the 18 °C on the night of 27 April) from narrow crevices in the cliff front. While the pitch shift may more easily be explained by the higher temperatures rather than by individual variations or technogenic effects, we cannot exclude that those fissures may both provide thermal conditions different from those in the open field and spatial constraints that may result in an alteration ("sound box effect") of the song, with selective amplification and damping of specific frequency bands as reported by Brizio et al. [15] in the case of *Eumodicogryllus bordigalensis* (Latreille, 1804) singing from close cavities.

Judging from the clarity of the harmonic structure and the volume of the 4th harmonic frequency, wider band recordings (e.g., at a sampling frequency of 96 or more kHz) may reveal high-order harmonic bands well above the audible range [15].

Initial attempts to identify song affinities from bioacoustic evidence only were unfocused and fruitless, as described in the Introduction. The attribution to the genus *Acheta* emerged only after the first pictures became available: subsequently, similarities with congeneric songs were investigated, starting with the species reported for the Italian territory.

*Acheta domesticus* (Linnaeus, 1758). The following observations are based on Massa et al. [1] and the accompanying DVD. Its song consists of a series of melodious "kre"-sounds (echemes), repeated more or less regularly at the rate of about 1–3/s. Echemes vary in the number of syllables they contain, between 2 and 4. Males produce a courtship song in the vicinity of a female. It consists of a continuous rustling mixed with loud, high-pitched ticks. Both the fundamental frequency, in the range of 4700/4800 Hz, the perceived timbre, and the harmonic structure of *A. domesticus* songs, with evenly spaced and well-structured harmonic bands, bear resemblance to the song of the n. sp., but the song structure is radically different, both in the calling/advertising song and in the courtship song.

*Acheta gossypii* (Costa, 1856). In Italy, the actual presence of this species is under scrutiny for the lack of type material and reliable reports since the original description. No bioacoustic description of this species exists.

*Acheta hispanicus* Rambur, 1838. In Italy, the species is reported from Calabria and Sicily. No song recordings from the Italian territory are available; the following observations are based on an audio sample from OSF [16] based on a laboratory recording from India (Uttar Pradesh, Aligarh). Song of *A. hispanicus* is made by very irregular bouts of 3–4 syllables echemes, reminiscent of the calling song of *Gryllus bimaculatus*, that are emitted at a rate of 1–6/s. The observed fundamental is at around 4500 Hz, the first harmonic band at around 9000 Hz is almost unnoticeable in the spectrogram and the frequency/sound pressure analysis, while the second harmonic is relevant at 14.5 kHz. The song of *Acheta pantescus* n. sp. is radically different both from the structural and the harmonic point of view.

*Acheta meridionalis* (Uvarov, 1921). Search for audio files attributed to this unusually widespread species (whose range spans from Macaronesia to Sudan) was fruitless and promoted direct contact with Axel Hochkirch, who studied *A. meridionalis* in the Canary Islands. After receiving our recordings and after comparing them with his own unpublished audio of *A. meridionalis* and *A. hispanicus*, A. Hochkirch (pers. comm.) confirmed that the song from Pantelleria didn't match either.

Also, other candidate species with Mediterranean and North African distribution were considered: *Acheta angustiusculus* Gorochov, 1993; *Acheta arabicus* Gorochov, 1993; *Acheta confalonierii* (Capra, 1929); *Acheta latiusculus* Gorochov, 1993; *Acheta rufopictus* Uvarov, 1957; *Acheta turcomanoides* Gorochov, 1993. Unfortunately, neither Capra [17]

nor Uvarov [18], Uvarov & Popov [19], nor Gorochov [3] included any bioacoustic description of these species, and the present authors could not locate relevant audio samples in the online repositories.

### 3.2.4. Material Examined

Italy, Sicily, Pantelleria Is., Lighthouse of Punta Limarsi 14, May 2022, C. Cusimano, P. Fontana, B. Massa (♂holotypus) (BMPC); Italy, Sicily, Pantelleria Is., Lighthouse of Punta Limarsi, 14 May 2022, C. Cusimano, P. Fontana, B. Massa (♀nymph, reared to adult paratype in alcohol) (MSNR); Italy, Sicily, Pantelleria Is., Martingana, 15 May 2022, C. Cusimano, P. Fontana, B. Massa (♀paratypus) (BMPC).

Other species examined: *Acheta turcomanoides* Gorochov, 1993. Algeria, Sahara, Hoggar (1♂) (MNCN); United Arab Emirates, Sharjah × Khor Kalba, 31 May–7 June 2006 (2♂♂„ 1♀) (BMPC); *Acheta confalonierii* (Capra, 1929). Libya, Cyrenaica, Porto Bardia, April 1927 (♂holotypus) (MSNG); the United Arab Emirates, Wadi Shawkah, 31 October–27 November 2006 (1♂„ 2♀♀) (BMPC); *Acheta hispanicus* Rambur, 1838. Spain, Siviglia, 16 September 1983, T. La Mantia (1♂„ 1♀) (BMPC); *Acheta rufopictus* Uvarov, 1957. Yemen, Socotra, Wadi Ayev, 10 April 2008, B. Massa (1♀) (BMPC); *Acheta domesticus* (Linnaeus, 1758). Italy, Sicily, Pantelleria Is. Loc. Mursia, 18 September 1993 (1♀), 9 September 1994 (1♂); Italy, Lombardy, Piacenza, 24 June 1989, B. Massa (1♀); Greece, Rodes, River Lutani, 28 July 1993, B.Massa (1♂); France, Corsica, Porto Vecchio, 30 August 1995, B. Massa (1♂„ 1♀) (BMPC); *Acheta meridionalis* (Uvarov, 1921) (photos of ♂and ♀from Canary Is. provided by A. Hochkirch).

### 3.2.5. Diagnosis

Small-sized, brachypterous with second wings atrophic, brown-reddish. Presently macropterous individuals are unknown, and those collected, as well as those observed singing in the field, were micropterous.

### 3.2.6. Description

Male. Small-sized for the genus. Color brown-reddish (Figures 4–6). Head round, uniformly brown, darker vertex with a transverse oval whitish spot between antennae, the width of the antennal cavity as large as the distance between these cavities (Figure 5b,c). Palpi pale brown, apically brown, a bit longer than the head length (Figure 5b), antennae pale brown. Pronotum weakly narrowed anteriorly, brown with a paler transversal band (Figure 5a). Tegmina straw-colored but with mostly dark veins, exceeding the apex of the 6th abdominal tergite, rather narrow, pale brown with a dark spot at the base of the dorsal part and a band along the upper edge of the lateral field (Figure 5a,d). Stridulatory vein short, three oblique veins plus a forth just outlined vein above, three transversal veins on the left of tegmen (Figures 5d,e and 8a), mirror transverse, medium-sized, short apical field of tegmina. Posterior wings are atrophic. Stridulatory file under the right tegmen consisting of ca. 240 teeth (Figure 6a) Legs uniformly pale brown, anterior tibiae with a large outer and a small inner tympanum. Fore tibiae with many hairs on upper and lower margins, 1 spur on the apical outer ventral tip, and 2 spurs on the apical inner ventral tip. Tarsi typical for the genus. Mid tibiae with 1 ventral and 1 dorsal apical spur on the inner (shorter) and on the outer margins (longer). Hind tibiae with 6 pairs of spines on dorsal margin + 2 spurs on each side, inner longer than outer ones. 1st hind basitarsomere with 6–7 pairs of very small dorsal spines + 2 apical spurs, inner longer than outer ones (Figure 6d). Last tergite with posterior margin straight, cerci as long as body (Figures 4 and 5a). Genitalia: posterior edge of epiphallus in dorsal and left lateral views as in Figures 6b,c and 9a.

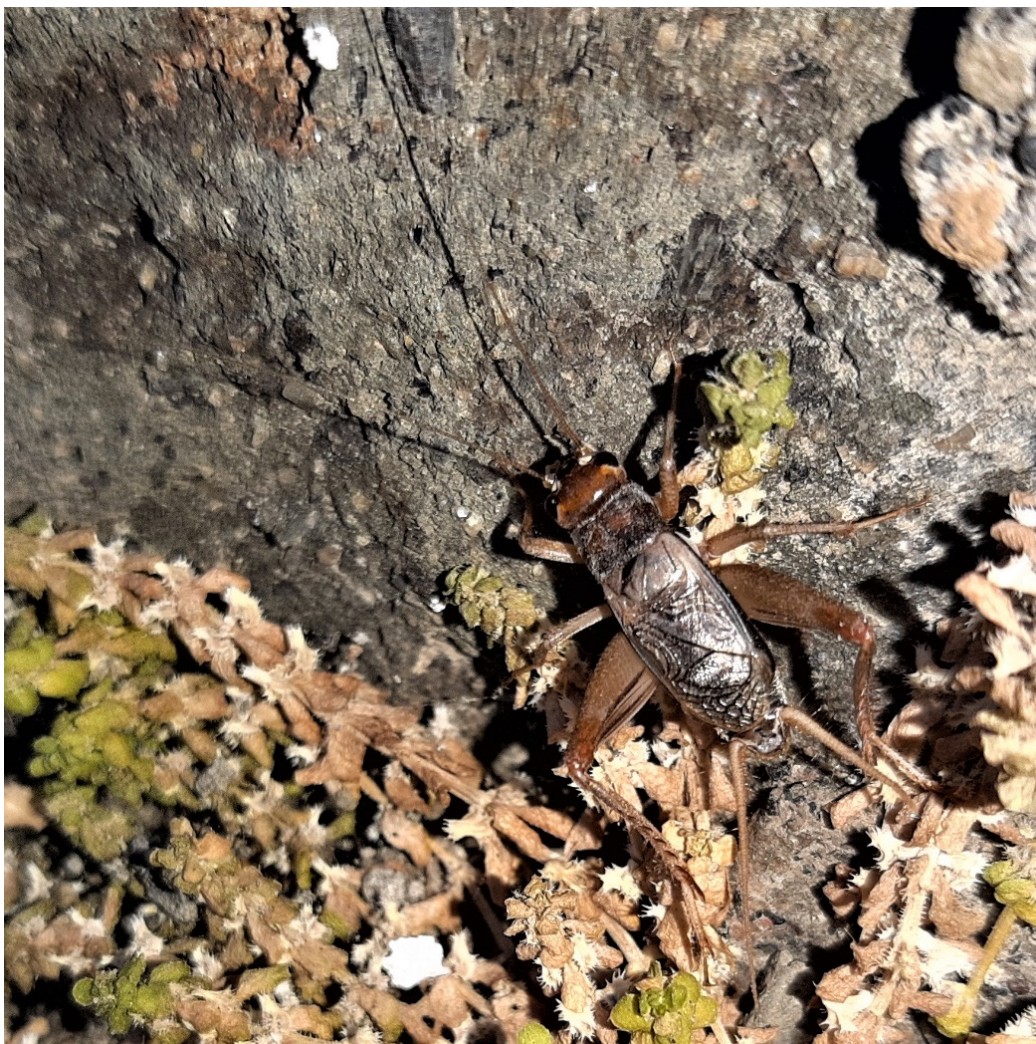

**Figure 4.** Live male specimen of Pantelleria Cricket *Acheta pantescus* n. sp. as photographed at the lighthouse of Punta Limarsi on 14 May 2022.

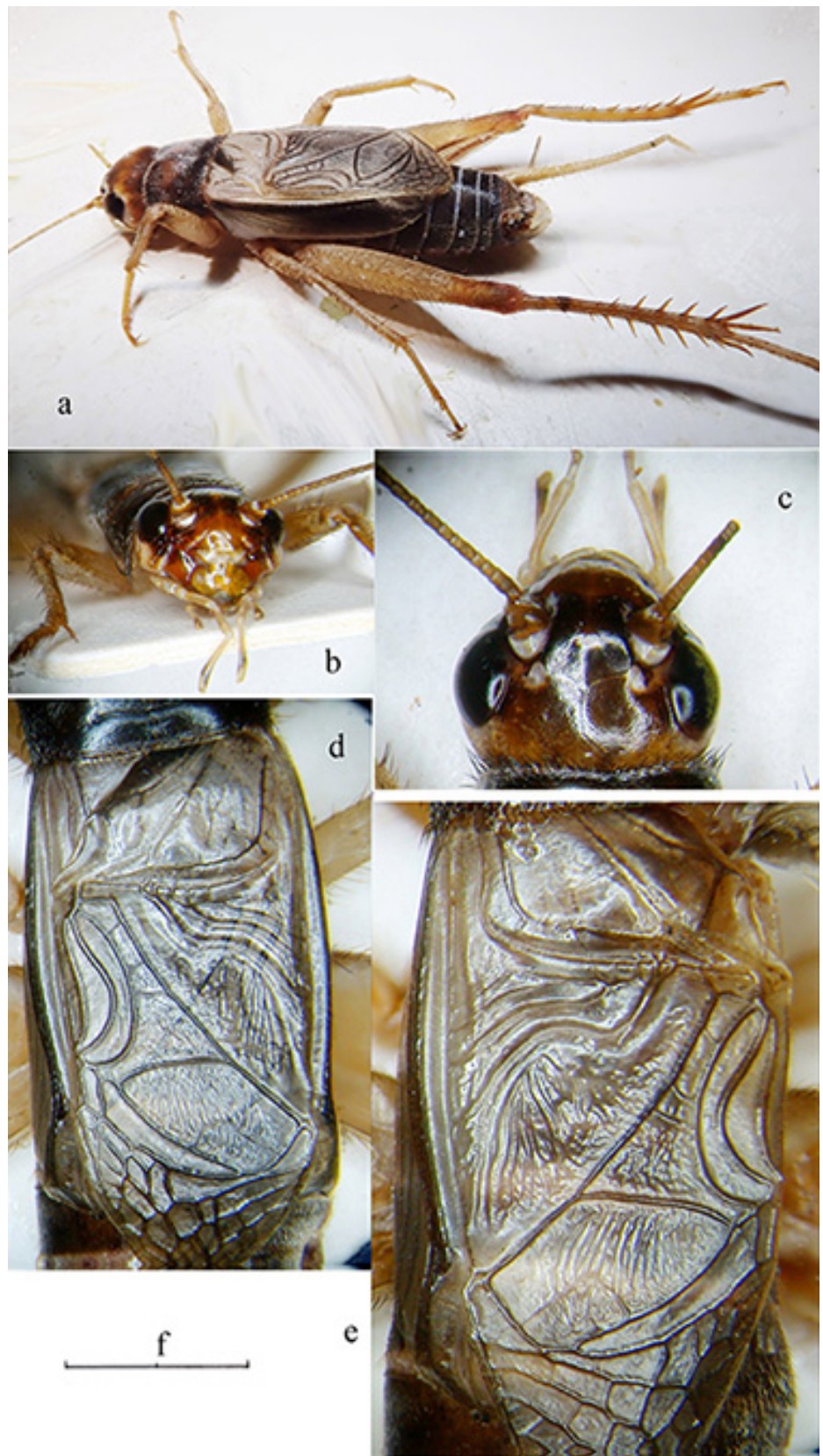

**Figure 5.** *Acheta pantescus* n. sp. male adult from the lighthouse of Punta Limarsi, Pantelleria. (**a**) lateral view; (**b**) head in frontal view; (**c**) head in dorsal view; (**d**) detail of male right tegmen; (**e**) detail of male left tegmen; (**f**) scale (a: 9.0 mm; b: 4.5 mm; c: 3.2 mm; d: 2.3 mm; e: 2.7 mm).

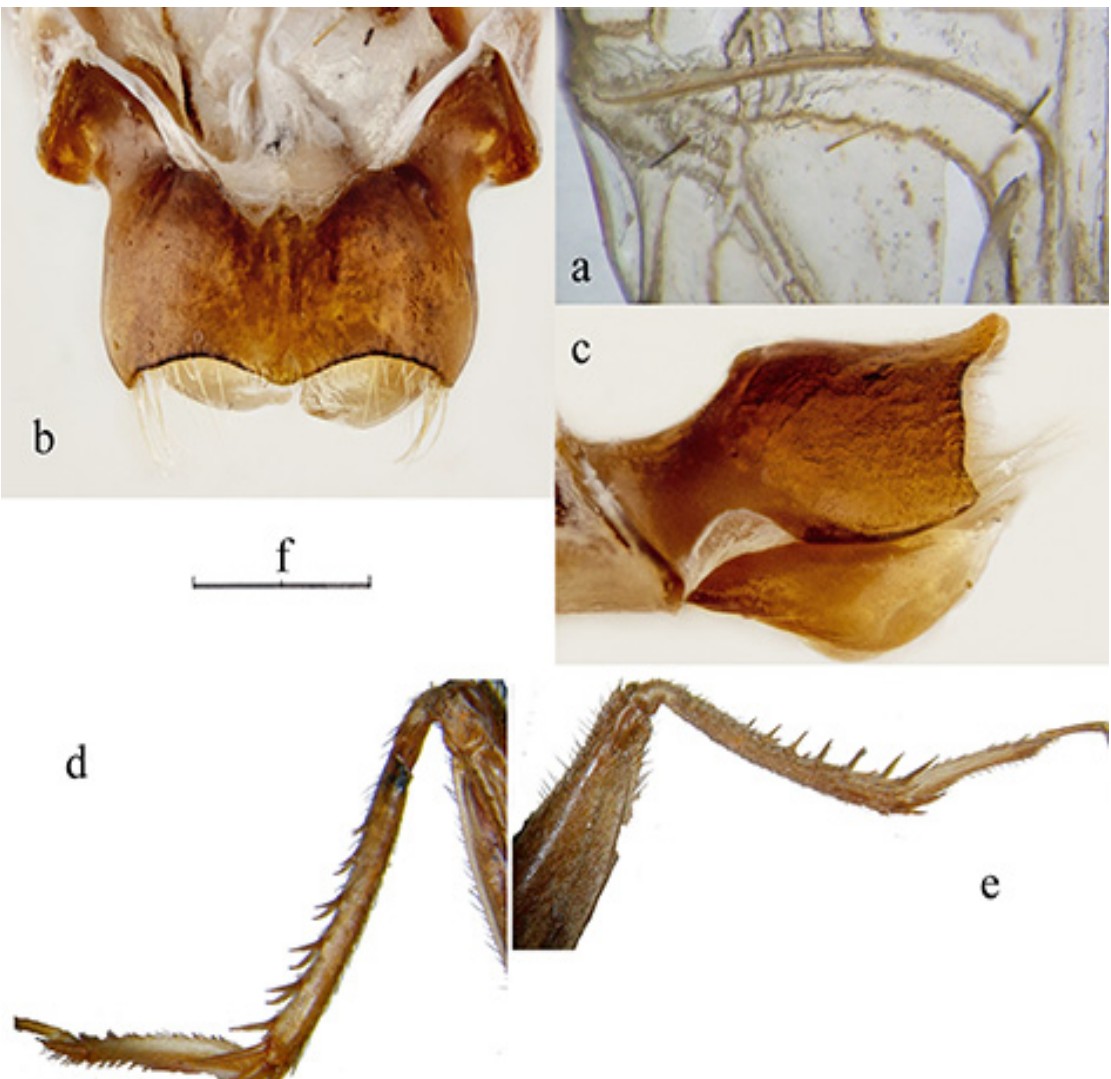

**Figure 6.** *Acheta pantescus* n. sp.; (**a**) stridulatory file under the right tegmen of male; (**b**) epiphallus in dorsal view; (**c**) epiphallus in lateral view; (**d**) hind tarsus and 1st hind basitarsomere of the male; (**e**) hind tarsus and 1st hind basitarsomere of the female; (**f**) scale (a: 1.3 mm; b: 1.0 mm; c: 0.6 mm; d, e: 7.0 mm).

Female. The same characteristics as the male, with the following differences (Figure 7a–d). Wider pale transversal band on pronotum. Tegmina just exceeding the 6th abdominal tergite (Figure 7a,c), dorsal part of tegmina pale brown. Legs with the same characteristics as the male (Figure 6e). Ovipositor slender (Figure 7a), the ventral valve of the ovipositor with a dozen dorsal very small spines (Figure 7e). Cerci are incomplete in both specimens.

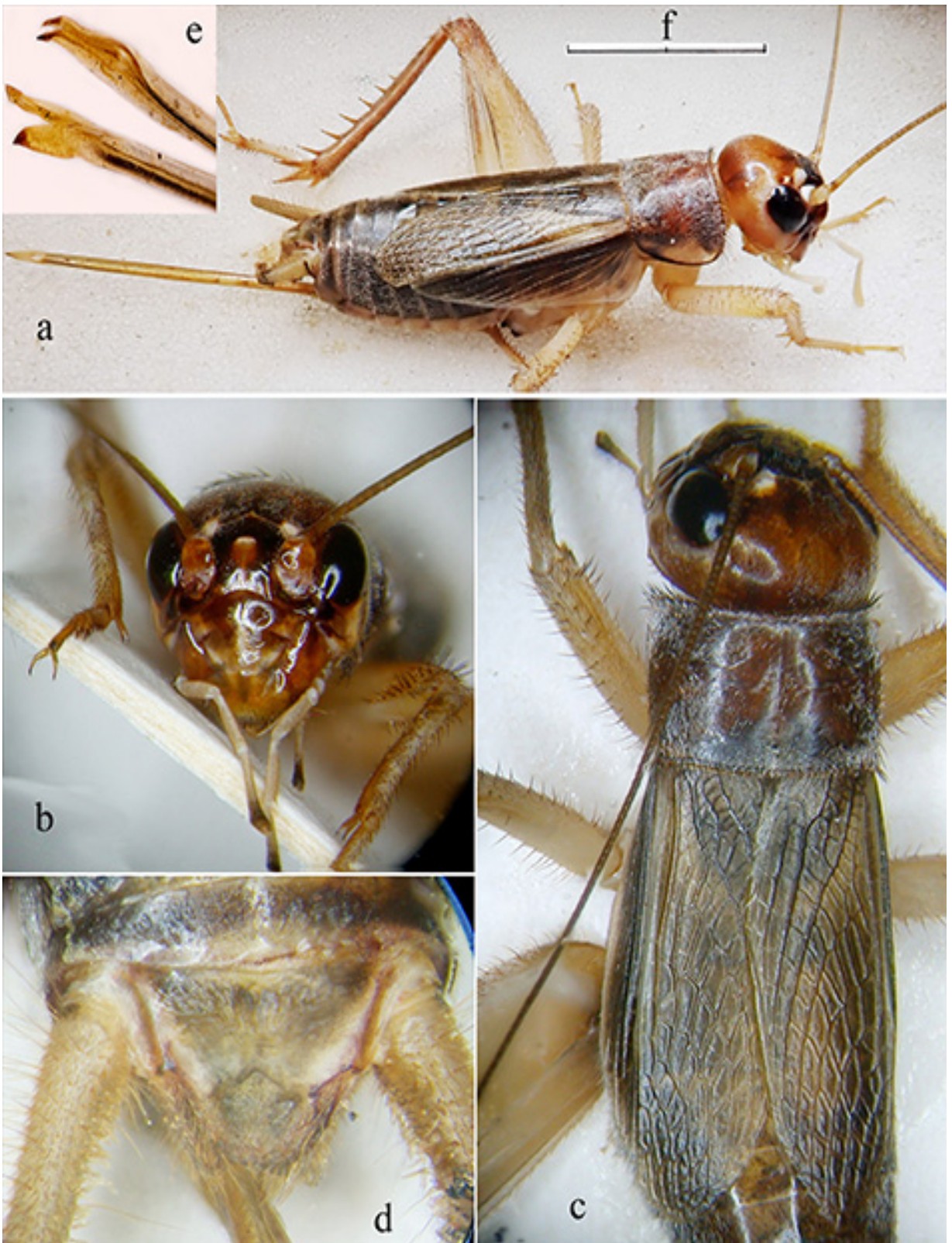

**Figure 7.** Adult female characters of *Acheta pantescus* n. sp. (**a**) lateral view; (**b**) head in frontal view; (**c**) dorsal view of head, pronotum and tegmina; (**d**) last tergite; (**e**) apex of ovipositor valves; (**f**) scale (a: 7.3 mm; b: 4.2 mm; c: 3.8 mm; d: 3.5 mm; e: 1.0 mm).

### 3.2.7. Measurements

Length of the body: 18.2 (♂), 18.3 (♀); length of pronotum: 2.5 (♂), 2.5 (♀); length of tegmina: 9.2 (♂), 8.0 (♀); length of hind femora: 9.7 (♂), 10.2 (♀); ovipositor: 12.2.

### 3.2.8. Etymology

The specific name *pantescus* is derived from the Latinization of the Italian adjective "pantesco", from the Sicilian language "pantisku", a clipping of "pantiddarisku", based on the island name, "Pantiddaria"—itself a derivation from the Arabic "Daughter of the winds", usually transliterated as *Bint al-riyāḥ*.

### 3.2.9. Affinities

A shortlist of related species was selected for comparison: *Acheta domesticus* (Linnaeus, 1758), widespread in the Palaearctic and North America; *Acheta gossypii* (Costa, 1856), presently known only from south Italy; *Acheta hispanicus* Rambur, 1839, known from Iberian peninsula, south Italy and North Africa; *Acheta meridionalis* (Uvarov, 1921), known from Sudan, North Africa to Iran and Canary Islands; *Acheta turcomanoides* (Gorochov, 1993) from Saudi Arabia; *Acheta confalonierii* (Capra, 1929) from Libya and Arabian peninsula; *Acheta arabicus* (Gorochov, 1993) from Saudi Arabia; *Acheta angustiusculus* (Gorochov, 1993) from Saudi Arabia; *Acheta latiusculus* (Gorochov, 1993) from Saudi Arabia; *Acheta rufopictus* (Uvarov, 1957) from Socotra.

Some of the previous species may have macropterous and micropterous forms. Others (e.g., *A. gossypii*) are known only from the micropterous form and only from the original description.

Considering the different quality, level of detail, and nature (line drawings or photographs) of the reference images from the original sources, included in Figures 6 and 7, we decided to avoid any interpretation errors that may derive from redrawing or tracing the original pictures, that are provided as they appear in the relevant sources.

Figure 8a–i show the male tegmina of the most closely related *Acheta* species.

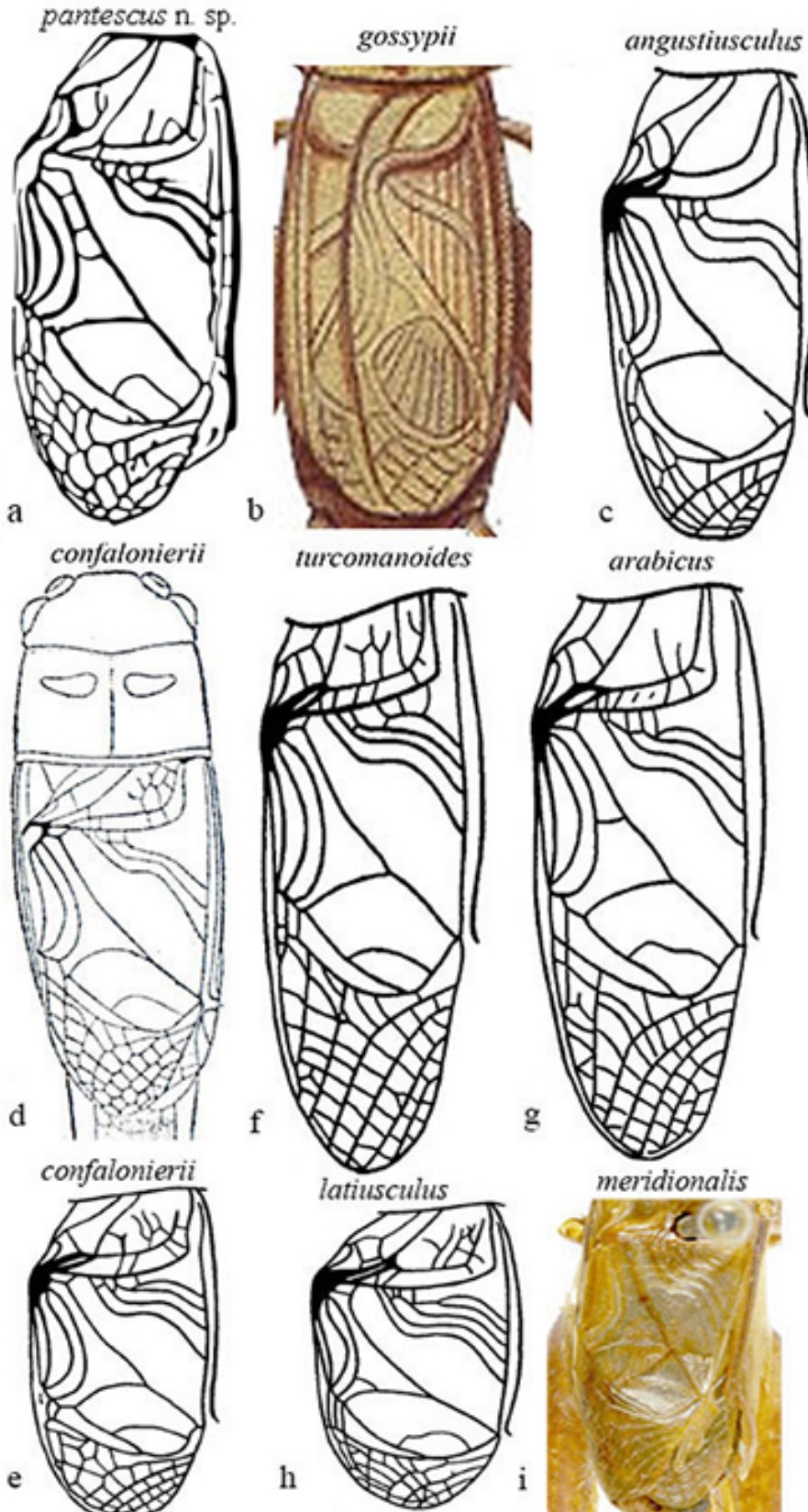

**Figure 8.** Right tegmen of eight species of Mediterranean *Acheta*. (**a**) *Acheta pantescus* n. sp., contrast-enhanced image; (**b**) *A. gossypii* (after Costa [20]); (**c**) *A. angustiusculus* (after Gorochov [3]); (**d**) *A. confalonierii* (after Capra [17]); (**e**) *A. confalonierii* (after Gorochov [3]); (**f**) *A. turcomanoides* (after Gorochov [3]); (**g**) *A. arabicus* (after Gorochov [3]); (**h**) *A. latiusculus* (after Gorochov [3]);

(**i**) *A. meridionalis* (holotypus of its synonym *A. canariensis*, after OSF online [16]) (with permission by the authors and by the Muséum National d'Histoire Naturelle of Paris).

Differences are evident between *Acheta pantescus* n. sp. (Figure 8a), *A. turcomanoides* (Figure 8f), *A. arabicus* (Figure 8g), and *A. meridionalis* (Figure 8i), which have four oblique veins [21]. *A. meridionalis* is micropterous, especially the female (as *Acheta pantescus* n. sp.), is blackish with a frontal yellow line on the head (specimens from Pantelleria were compared with two photographs of *A. meridionalis* from Canary Is. provided by A. Hochkirch). *A. angustiusculus* (Figure 8c) and *A. confalonierii* (Figure 8d,e) have three oblique veins (macropterous form) and four oblique veins (a micropterous form of the latter), but lack transversal veinlets on the left area of tegmen. *A. gossypii* (Figure 8b), as schematized by Costa [20], has only two oblique veins and lacks transversal veinlets. *Acheta domesticus* and *A. hispanicus* may be excluded, other than by right tegmen veinlets, and also due to the different song [1,16] ( see above).

Figure 9 shows the genitalia in dorsal and lateral view, respectively of *Acheta pantescus* n. sp. (Figure 9a), *A. confalonierii* (Figure 9b), *A. turcomanoides* (Figure 9c), *A. angustiusculus* (Figure 9d), *A. arabicus* (Figure 9e) and *A. meridionalis* (Figure 9f).

Consulting the paper by Gorochov [3], in particular comparing the genitalia (Figure 9), the most closely related species seems to be *A. confalonierii* from Libya (Capra) [17], also found in the Arabian peninsula, which is bigger; the latter has different lateral lobes of the epiphallus, which rather resemble those of *A. turcomanoides*. However, there are two or three species of *Acheta* very similar to *A. confalonierii* in North Africa and the Arabian peninsula, differing only in the male genitalia (A. Gorochov, V. Llorente, pers. comm.) [3].

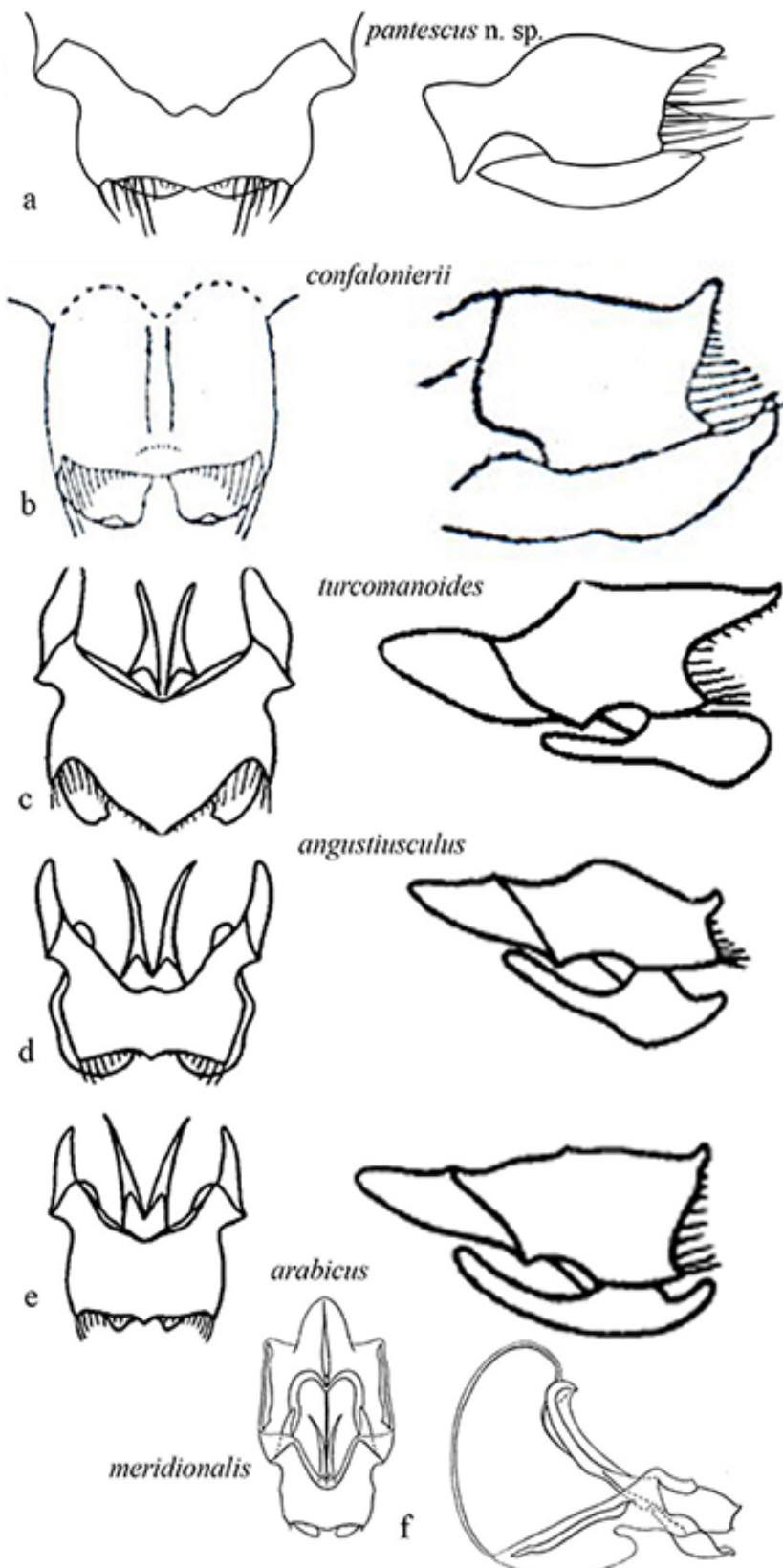

**Figure 9.** Genitalia of some Mediterranean species of *Acheta*. On the left epiphallus and ectoparameres from above, on the right is the left lateral view: (**a**) *Acheta pantescus* n. sp.; (**b**) *A. confalonierii* (after Capra) [17]; (**c**) *A. turcomanoides* (after Gorochov) [3]; (**d**) *A. angustiusculus* (after Gorochov) [3]; (**e**) *A. arabicus* (after Gorochov) [3]; (**f**) *A. meridionalis* (after Gorochov & Llorente) [22] (with permission by the authors).

### 3.2.10. Distribution and Conservation Assessment

Pantelleria Cricket *Acheta pantescus* n. sp. is present in the Italian territory, and is currently reported from a few stations on the Island of Pantelleria (Figure 10).

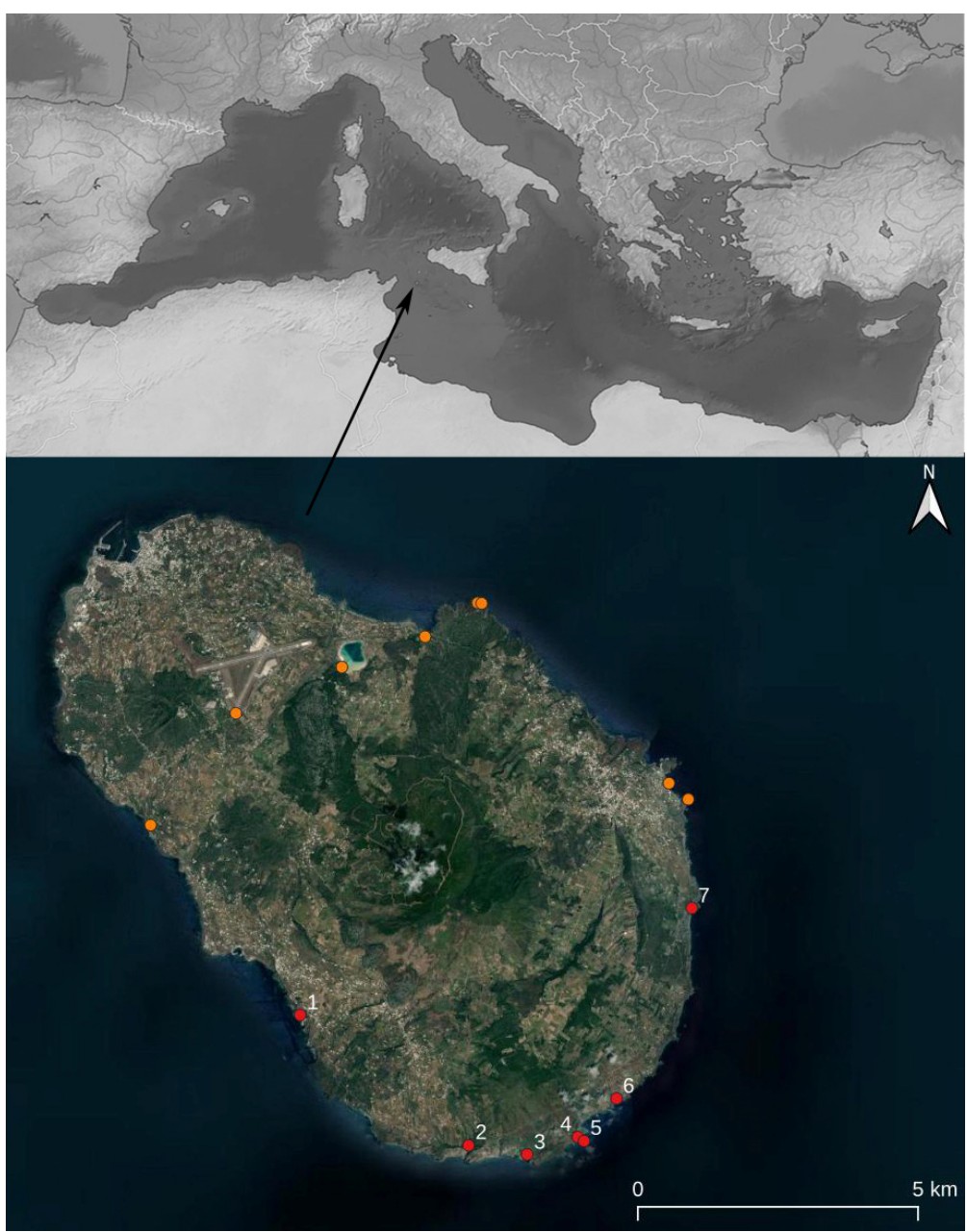

**Figure 10. (Above)**: Geographical position of the Island of Pantelleria in the Mediterranean. (**Below**): localities of Pantelleria where night recordings have been carried out. Red Dots mark stations where the new species was recorded, and orange dots where it was not recorded. See legend in Table 1.

The intrinsic vulnerability of insular habitats under increasing anthropic pressure requires some form of a conservation initiative, such as its inclusion in a special Red List by the IUCN Italian Committee, aimed at ensuring the survival of this newly-reported, unique, and elusive species of cricket. Presently *Acheta pantescus* n. sp. has the following distribution criteria: Extent of occurrence EOO: less than 100 km$^2$; Area of occupancy AOO: less than 10 km$^2$. However, based on the available information, it would better fit under the Vulnerable category (Criterion D2).

### 3.2.11. Habitat on Pantelleria Island

During the survey to detect the presence of seabirds on Pantelleria, recorders were placed in further sites on the island coast, but the presence of the cricket was confirmed only in those marked by a red dot in Figure 10.

The peculiar song of the new species has never been heard in the inland part of the island during several research projects carried out by PF in 2021 and early 2022.

*Acheta pantescus* n. sp. has been found only on the south and southwest coasts of Pantelleria, between the localities of Scauri and Punta del Formaggio (Figure 10), consisting of steep and indented cliffs. Their typical lithology is volcanic, consisting of various overlapped lava layers attributed to the "pre-Green Tuff lavas and tephra" (as clarified in the Discussion). The cliffs are subject to frequent collapses and landslides and vary in height, reaching altitudes of up to 200–300 m overlooking the sea. This coastal area, from the point of view of bioclimatic levels, falls between the semi-arid infra-Mediterranean level and the dry Mediterranean thermal level [23]. On the coastal cliffs, in the lower parts most exposed to storm surges, a more or less sparse halo-rupicolous coenosis is found in *Limonietum cosyrensis*. Moving to the higher parts, sheltered from the sea but not from the salt, pioneer vegetation is represented by coenosis of subalophilic chamaephites *Matthiolo pulchellae-Helichrysetum errerae* and *Silenus sedoidis-Bellietum minuti*. On the plateau developing on the top of the cliffs, a vegetation of rocky substrates and sub-coastal areas of high scrub (*Periploco angustifoliae-Juniperetum turbinatae*) and low scrub (*Periploco angustifoliae-Euphorbietum dendroidis* and *Genisto aspalathoidis-Rosmaniretum officinalis*) can be observed, with fragmentary aspects of *Opuntia ficus-indica*, the latter introduced in the past by man as an agricultural crop.

*Acheta pantescus* n. sp. lives mainly among the fractures of the lava rocks a few meters above sea level and in the steep sandy sides, from which, well hidden, the males emit their song at night. Figures 11–13 show some habitat types where the cricket has been found.

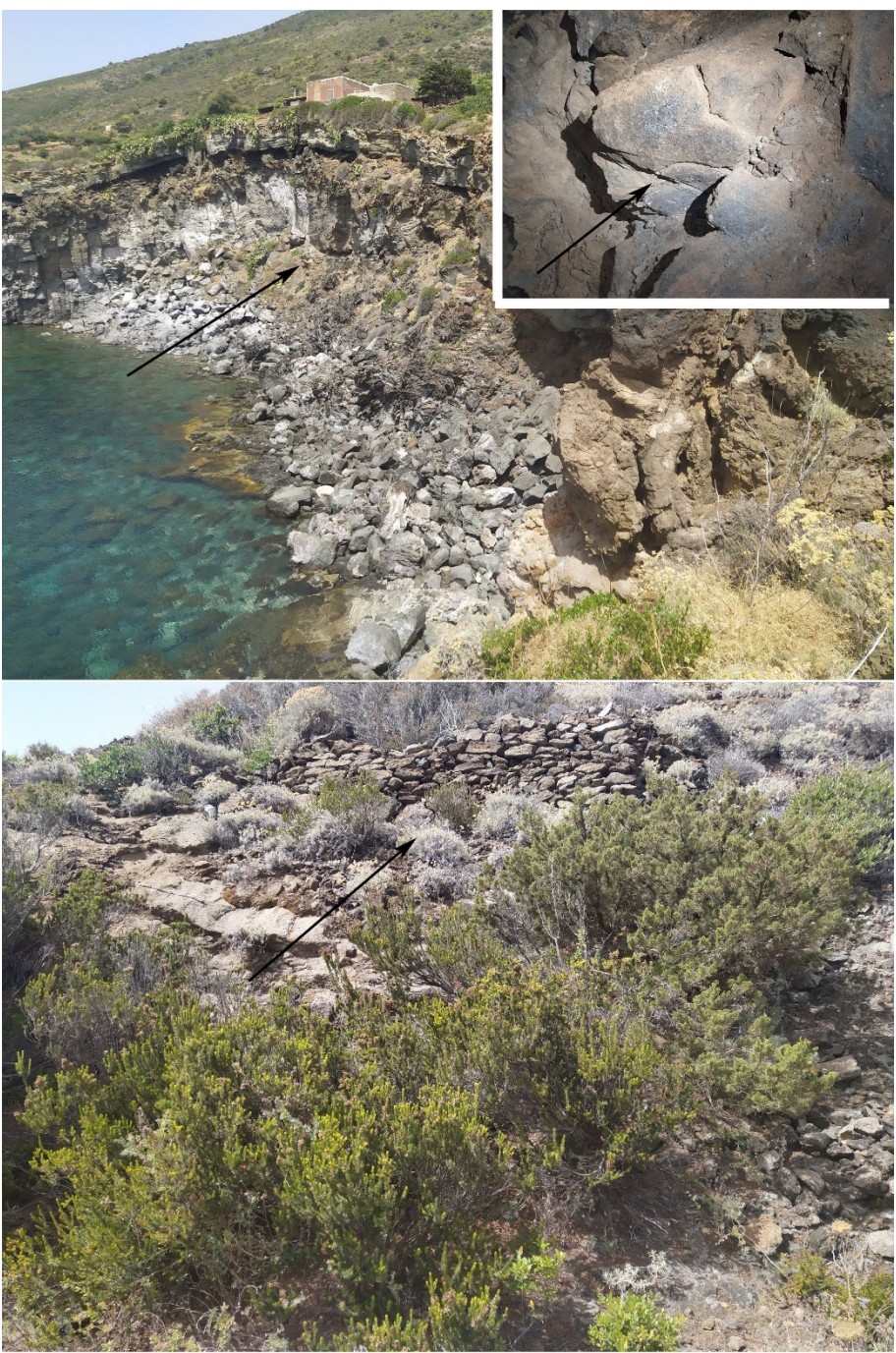

**Figure 11. (Above**): Cala Rotonda, Martingana; the arrows indicate the area inhabited by *Acheta pantescus* n. sp.; in the inset, a volcanic rock photographed at night; the cricket sang inside the crevice (see arrow). (**Below**): *Juniperus turbinata*; the cricket on the night of 6 July 2022 was singing behind this plant, and its recovery was the dry stone wall.

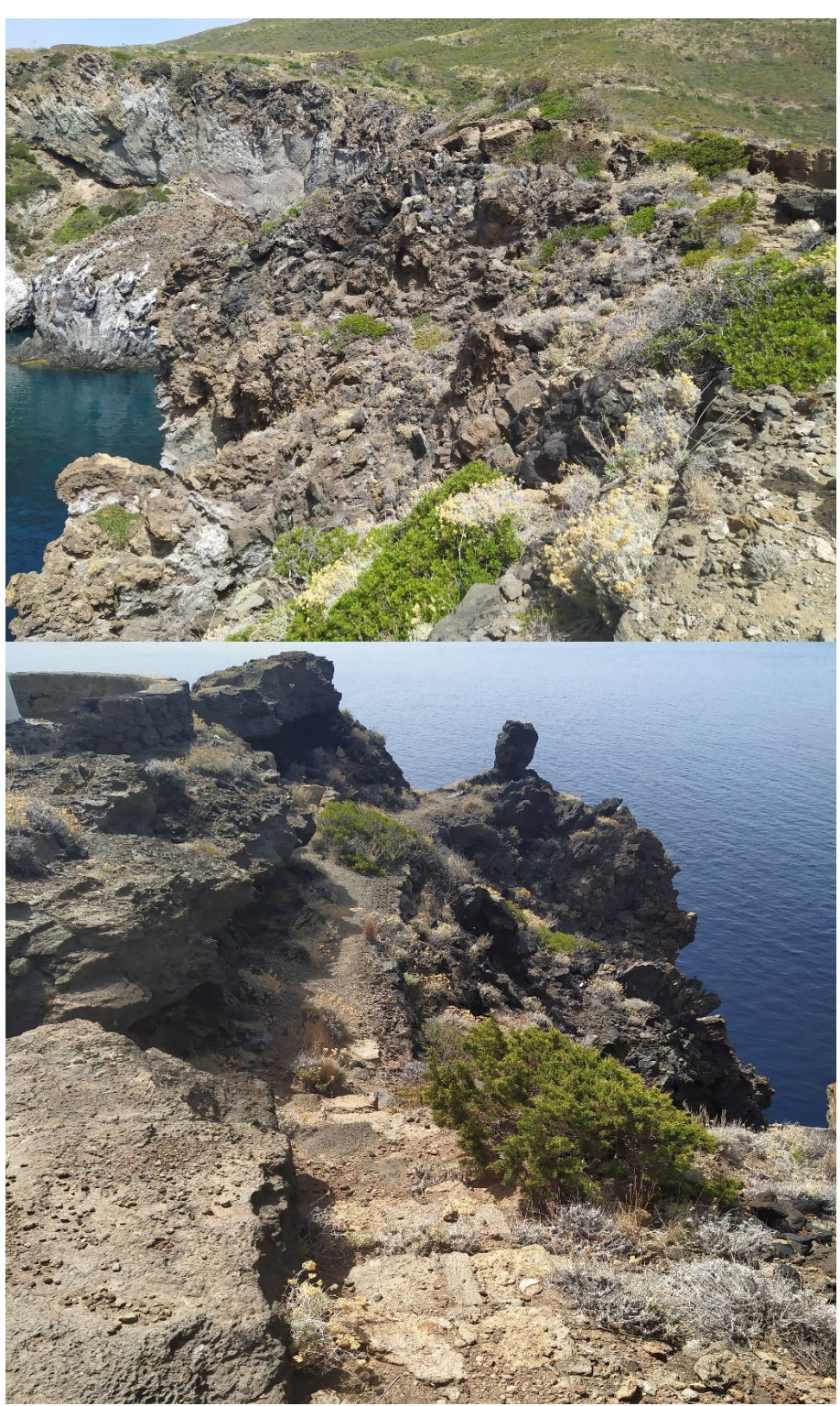

**Figure 12.** Habitat of *Acheta pantescus* n. sp. in the locality Punta Limarsi, next to the lighthouse.

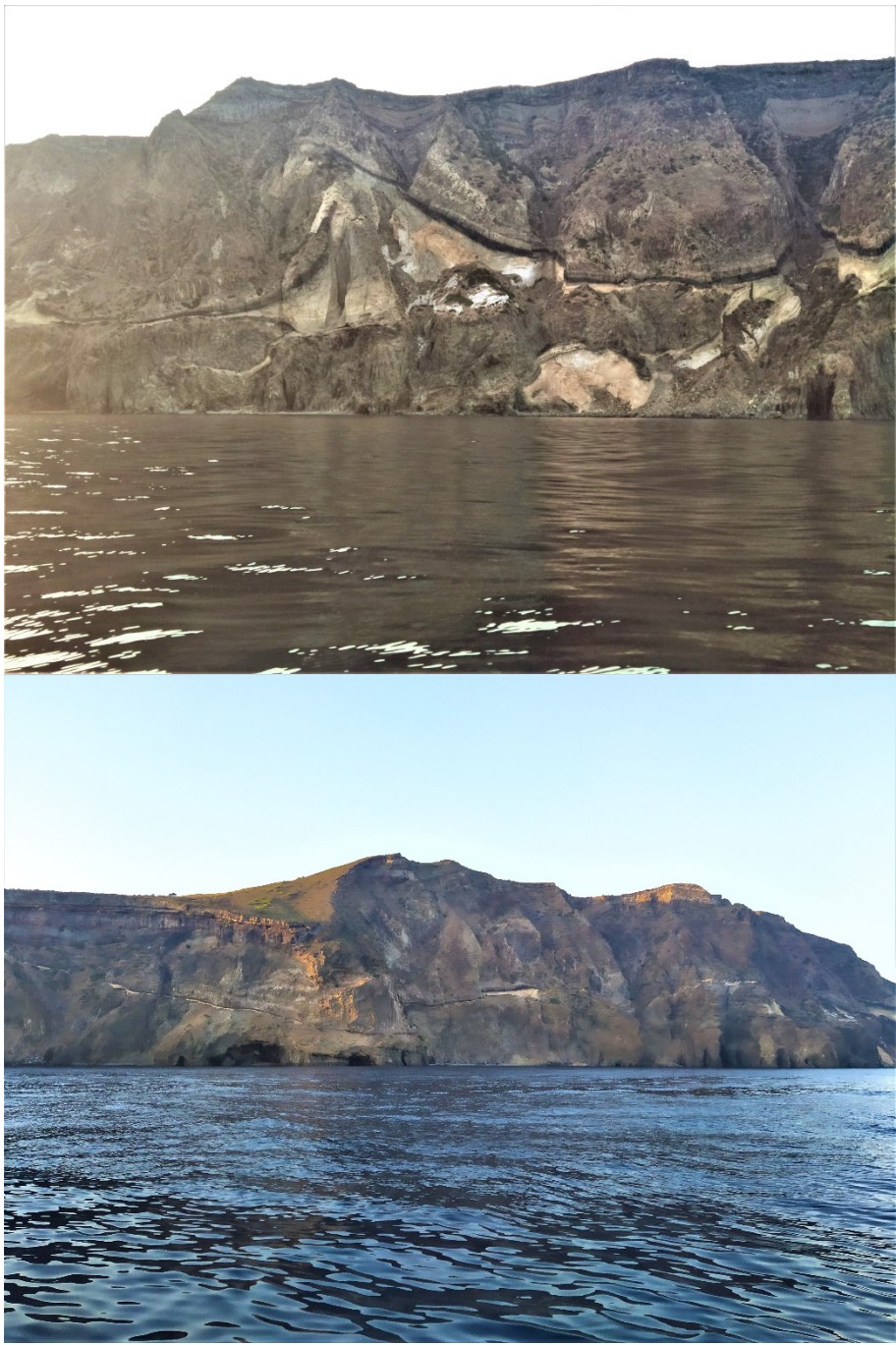

**Figure 13.** Two aspects from the sea of Punta Limarsi (Pantelleria Is., Sicily), where the newly discovered *Acheta* lives.

## 4. Discussion

If an extensive set of DNA sequences of Mediterranean congenerics was available, the DNA sequencing of the new species could help to elucidate its phylogeny; unfortunately, only *A. domesticus* and *A. rufopictus* are covered by GenBank. For that reason, genetic investigations were omitted.

In accordance with Massa [24], the presence at Pantelleria of phytophagous species strictly linked to allochthonous plants imported to the island in the last decades is certainly proof of a speedy immigration of insects, at least concerning small and light species driven by the winds. However, the present fauna and flora of Pantelleria are more related to their Italian than to their North African counterparts [1,24–28].

Presently 12 species of Grylloidea are known from Pantelleria [24,29]: *Gryllus bimaculatus* De Geer, 1773; *Acheta domesticus* (Linnaeus, 1758); *Eumodicogryllus bordigalensis* (Latreille, 1804); *Arachnocephalus vestitus* (Costa, 1855); *Trigonidium cicindeloides* (Rambur, 1838); *Pseudomogoplistes squamiger* (Fischer, 1853); *Myrmecophilus baronii* (Baccetti, 1966); *Myrmecophilus ochraceus* (Fischer, 1853); *Oecanthus pellucens* (Scopoli, 1763); *Gryllotalpa cossyrensis* (Baccetti & Capra, 1978). To them we add also *Oecanthus dulcisonans* (Gorochov, 1993) (B. Massa, pers. obs.). All the previous species are generally widespread in the Mediterranean.

*Acheta pantescus* n. sp. is an unexpected taxon not only for Pantelleria but also for the Mediterranean area, which is well explored and studied from the orthopterological point of view. This unreported species escaped detection despite its peculiar song because of the unfrequented and impervious terrain where it lives: rocks and steep slopes only a few meters above sea level on the volcanic island of Pantelleria. Very likely palaeogeographical dynamics influenced the degree of connectivity in the Mediterranean during the Pleistocene, when fluctuating eustatic sea levels, a result of the alternation of glacial and interglacial phases, led to intermittent. Still, prolonged isolation of species as islands formed during sea-level highstands and, conversely, allowed species influxes across previously submerged regions as natural land bridges formed terrestrial linkages during lowstands [30]. The most recent period of Quaternary glaciation or Last Glacial Period (between approximately 115,000 to 11,700 BP, peaking during the Last Glacial Maximum (LGM) between 26,500 and 15–18,000 years ago) was one such significant event thought to have facilitated the exchange of flora and fauna between Sicily and Malta [31,32]. Pantelleria was the theater of a Plinian eruption 45,000 years ago, with the deposition of the strongly peralkaline air-fall Green Tuff; thus, it is usually believed that no fauna and flora remained on the island (Silvio Rotolo, pers.comm.). Anyway, it should be noted that the observed distribution of *Acheta pantescus* n. sp. seems to concentrate or to be restricted to "pre-Green Tuff lavas and tephra" cliff faces (see maps in [33,34]), not covered by the Green Tuff deposition. In line with principle, those cliffs could have provided refugia from the Plinian eruption in case of an ingression of the new species earlier than 45,000 years ago. However, according to Lodolo & Ben-Avraham [35], during the LGM, the Adventure Plateau was part of the former Sicily mainland, forming a peninsula (the Adventure Peninsula). Lodolo & Ben-Avraham believe that most likely the ancient inhabitants of the Adventure Peninsula came from Sicily, with which a direct terrestrial connection existed throughout the LGM, as indicated by morphological reconstructions of paleo-shorelines (Figure 14). The provenance from North Africa would have been more difficult because of a nearly 50 km wide seaway between the peninsula and the former Tunisian shore.

One of the widespread species of the genus is *Acheta domesticus*, a medium-sized light fulvous or testaceous cricket whose pronotum is adorned with two large brown spots and whose lateral lobes are testaceous with a brown band in the superior part. This species, geographically widespread and accidentally introduced in many countries, often synanthropic and today widely bred as live food for various animals, is generally macropterous and has 4–5 oblique veins on the right tegmen. When Costa described *Gryllus gossypii* [21], which infested cotton cultivations in the area of Otranto (Apulia, south Italy), he highlighted the possibility that it was a brachypterous form of *A. domesticus*. Regrettably, we were not able to find the types which had to be preserved in the Museum of Zoology of Naples University, and *A. gossypii* was no more collected since its description. Indeed, the female, identified as *A. gossypii*, collected in Murcia (Spain), probably belongs to another species [22]. Concerning this species, Chopard [35] also reported it in Kenya, but this record has not been supported by specimen collection.

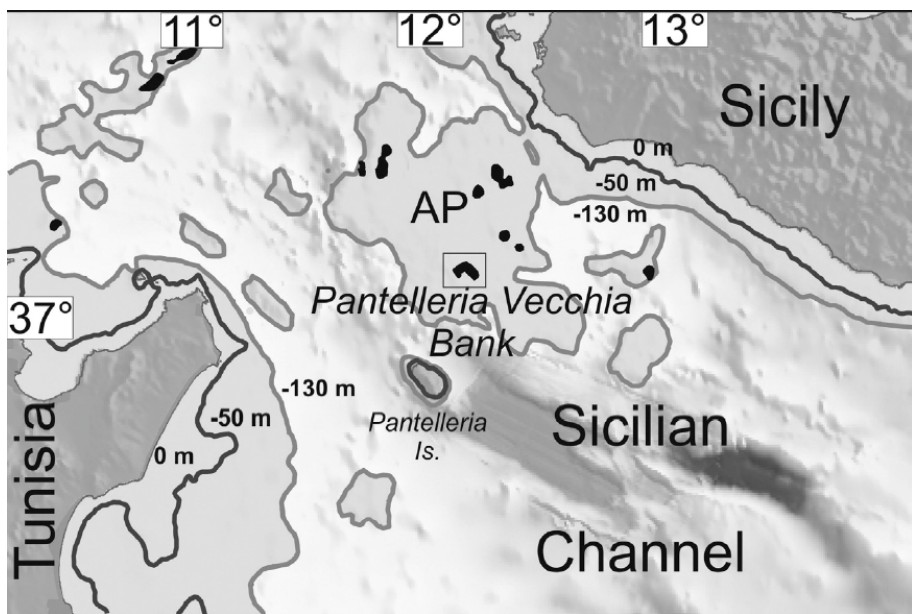

**Figure 14.** Adventure Plateau dominated by north-west trending, linked to Sicily during the LGM and a few km away from the island of Pantelleria (after Lodolo & Ben-Avraham [28]) (with permission by the authors).

Thus, we may propose three hypotheses. The first is that the newly discovered *Acheta* from Pantelleria arrived from Europe (Sicily during the LGM was only a few km away from Pantelleria [36]); in this case, the only possible colonization was passive, that is, the spread of propagules by possible means, such as floating logs or even the wind. These propagules likely settled along the coast in a very peculiar habitat, where they lost their second pair of wings, became micropterous, lost the ability to fly, and underwent size reduction, becoming endemic to the island. This may have happened during the last 15–26,500 years. Colonizers very likely remained linked to the coasts of Pantelleria.

The second hypothesis is that the propagules arrived from North Africa, which during the LGM remained at least 50 km away from Pantelleria. However, according to Thiel & Gutow [37], Orthoptera are not on the list of rafting organisms, like Coleoptera, Hymenoptera, Diptera, etc. Thus, the colonization from Sicily looks more probable than from North Africa. An interesting similar case concerns *Stenostoma cossyrense* Bologna, 1995 (Coleoptera: Oedemeridae), which, however, has a much-related species living in Malta and Sicily (*S. melitense*, M. Bologna, pers. comm.).

A third hypothesis could be that of transport by ancient human populations who have visited Pantelleria since prehistoric times, both as a stopover for navigation in the Sicilian Channel and for the precious deposits of Obsidian. According to Rapisarda [38], the exploitation of the obsidian deposits of Pantelleria could also have begun in the Paleolithic, starting 18,000 years ago, when the level of the Mediterranean was lower than 110 m. Therefore, navigation from North Africa to Sicily was practicable. *Acheta pantescus* n. sp. could therefore also be a species transported by man (for example, via artifacts, foodstuffs, or live plants) that then went extinct or was never found again in its place of origin, or a species that evolved from a small number of individuals accidentally introduced onto the island. All three hypotheses match the patterns described by Kenyeres et al. [39] with reference to the endemisms of the islands in the Mediterranean Sea.

In recent years, insular cricket and insect faunas have received much attention, e.g., by Otte et al. [40] and Hembry et al. [41]. *A. pantescus* n. sp. provided evidence of cryptic, elusive species even in a well-studied geographical area such as the Mediterranean. The authors hope that further investigation of the novel species from Pantelleria will provide the opportunity to reconsider the cricket fauna of the island from the more general point of view of biogeographic speciation patterns.

**Author Contributions:** Conceptualization, B.M., P.F. and C.A.C.; methodology, B.M. and P.F.; software, B.M. and C.B.; field expeditions, B.M., C.A.C. and P.F.; bioacoustic investigation, C.B.; editing, proofreading, bibliographic researches, all authors. All authors have read and agreed to the published version of the manuscript.

**Funding:** 'Study on the honey bee population of Pantelleria' by the Technological Transfer Center of the Edmund Mach Foundation and the Pantelleria Island National Park; 'Monitoring of Breeding Seabirds in the Sicilian Region' by the University of Palermo, Department SAAF on behalf of Istituto Superiore per la Protezione e la Ricerca Ambientale.

**Institutional Review Board Statement:** Not applicable.

**Data Availability Statement:** Not applicable.

**Acknowledgments:** The first report of *Acheta pantescus* n. sp. was possible thanks to activities carried out during the Monitoring of Breeding Seabirds in the Sicilian Region, co-ordinated by Tommaso La Mantia (University of Palermo, Department SAAF) on behalf of Istituto Superiore per la Protezione e la Ricerca Ambientale (ISPRA); we would thank T. La Mantia very much, who involved us in this research. The present publication was possible thanks to the Edmund Mach Foundation and the Pantelleria Island National Park. We are indebted to Marcello Romano, who photographed at high-resolution genitalia of the new species. We also thank Silvio Rotolo for his interesting information on the paleogeography of Pantelleria. The choice of the name of the new species of *Acheta* resulted from an opinion poll promoted by the Parco Nazionale di Pantelleria; we are indebted for this to Sonia Anelli and Andrea Biddittu. We also thank Marcello Romano and contributors to the forum entomologiitaliani.net very much. Further, we thank Andrej Gorochov, Axel Hochkirch, Vicenta Llorente and Emanuele Lodolo for the permission to use the drawings appearing on their papers cited in the references, as well as for the useful information they provided on morphology and songs of other Mediterranean species of *Acheta*, Laure Desutter-Grandcolas for the permission to use the photo of *Acheta meridionalis* preserved in the Muséum National d'Histoire Naturelle of Paris, Marco Bologna for information on the distribution of endemic *Stenostoma*, and Filippo Buzzetti for some data on conservation assessment. Sara Chiarello, Tommaso La Mantia, Rocco Lo Duca, and Marta Visentin shared with BM and CC the visit to Pantelleria in April 2022. Giovanni Bonomo and Denny Salvatore Almanza from Pantelleria gave us some useful information on Pantelleria. Finally, we thank the Servizio Informativo Agrometeorologico Siciliano, who provided data on temperature and humidity from Pantelleria.

**Conflicts of Interest:** The authors declare no conflict of interest.

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
