# Peer review of "New Unexpected Species of Acheta (Orthoptera, Gryllidae) from the Italian Volcanic Island of Pantelleria"

_diversity, doi:10.3390/d14100802_

Round 1

Reviewer 1 Report

In this paper, the authors describe a new species of the cricket genus Acheta from a small island near Sicily. The species was discovered by chance using acoustic recordings and specimens were looked for only in a second time. The authors describe the morphology of the species, give photos of male and female specimens, and describe the songs recorded at two different time in the year. They finally discuss the possible origin of the species.

This paper is interesting for several topics. It first shows the usefulness of acoustic recording in general biodiversity surveys. It illustrates also biodiversity richness, by the presence of  new species in such a small island in a well-studied area. So this paper must be published.

The formal description of the species should however be more extensive with more characters and illustrations.

The authors should describe more precisely the head (palpi), the legs (spurs and spines of tibiae I, II and III; hindbasitarsomeres), the stridulatory apparatus of the male (number of teeth of the stridulatory file), and the male and female genitalia.

Male genitalia are illustrated only by two photos which are very small and not of good quality: this cannot allow to represent properly the structures of the genitalia.

Female genitalia are not illustrated, while they are more and more used in cricket taxonomy and prove informative at genus and species level.

The photo of the cerci are not useful and could be deleted, and clearly replaced by photos of hindtibiae and larger photos of the genitalia.

The illustrations of the male genitalia and male forewing of related species should be redrown to be of better quality for comparison. Note that the lateral figure of male genitalia of A. meridionalis is reversed dorso-ventrally.

Scales should be given for all the figures.

For acoustics, the "recorded band" in Table 2 could be suppressed, as it is the normal relation for recorders.

The authors should give the temperature at the time of recording, as this variable is known to greatly influence the frequency of cricket calls. The difference between the recorded calls may be due to temperature difference  between april and july.

The reference to macropterous individuals (male description) should be suppressed, as only one male has been collected. The references to macropterus / micropterous species should also provide the number of individuals known per species, as polymorphism may not be detected with too few specimens.

I would recommand that the paper is published after major revision. The best would be adding DNA sequences, at least for COI and some frequently used markers in cricket taxonomy and phylogeny (12S, for example). It must be noted that sequences are already present in GenBank for Acheta domesticus and A. rufopictus.

Reviewer 2 Report

The paper by Massa, Cuismano, Fontana, and Brizio, which describes a surprising new species in a cosmopolitan, well-studied and well-sampled genus, is worthy of publication by this finding alone. Also of note is that this new species was discovered as a byproduct of passive acoustical monitoring for more charismatic species, showing broader impacts from taxon-focused conservation research. The new species hypothesis is convincingly reported despite small sample sizes for morphology and bioacoustics, and the absence of recordings under controlled conditions in the laboratory. This species is apparently difficult to collect! Relevant descriptions, diagnosis, bioacoustics, and ecology are included. Some suggestions to improve the acoustical data and terminology, and manuscript readability and conciseness follow.

This manuscript is written as a hybrid between a discovery narrative and a taxonomic work. The authors may consider choosing one format and ordering the manuscript accordingly. For example, the species was first discovered by acoustical monitoring, and by song alone the authors suspected the species was new. However, songs of related species are later shown to be scarce (“fruitless” research mentioned in line 44), so it is not clear how this initial new species hypothesis was made: presumably by eliminating all other likely grylline song possibilities. The process of elimination that led to hypothesis formation may be explained earlier in the manuscript, following the flow of the authors’ journey of discovery. Presumably, grylline possibilities other than Acheta were also considered, and none are not mentioned until the Discussion. Reflecting this, song descriptions and comparative acoustical analysis may appear first in Results, followed by morphology and ecology. This is a stylistic choice, but it may read more coherently either if it is a narrative or a traditional taxonomic paper layout.

Introduction

Nice narrative of discovery!

Plants and plant communities are mentioned in several places in the manuscript. It is standard to include taxon authorship at first mention of a plant taxon. Perhaps for the birds too in the Introduction?

Methods

Some Methods are actually Results. For example, the Fig. 1 map shows results of fieldwork and recording. These are properly Results obtained by implementing various Methods.

Include a reference for the acoustical terminology used. The authors presumably follow Ragge & Reynolds using echeme as a higher order sound unit. Recently updated and standardized in Baker and Chesmore 2020.

Consider combining Tables 1 and 2. Much of the information is redundant.

Temperature data are missing. Temperature is critical when measuring pulse, syllable, and echeme rates in crickets. Even frequency may be affected (e.g. Walker & Funk 2014). Passive acoustical monitoring stations may have ambient temperature available. If temperature was not taken, this must be noted when pulse rates are described later in the ms. Future authors may wish to compare their pulse and syllable rates with this work and cannot properly do so without a reference temperature.

Line 105: explain why recordings are not comparable. Presumably, the frequency domain may be affected by the limits of the recording equipment. Temporal features may be comparable.

Line 126: Was quantitative song analysis performed in Adobe Audition? How were pulse rates measured? Manually or automatically?

Line 134: Include reference for Blackman-Harris method.

Results

Section 3.1: crowdsourced choice of new species name may belong in Methods. The name chosen is a Result.

Line 201: am I correct in interpreting Fig. 5a as showing a partial 4th oblique vein in the n. sp.?

Line 202: my ms version does not show any arrows, which would greatly help visualizing the veination characters.

Line 248: Is A. confalonierii polymorphic for oblique vein count? Fig. 5d shows 3 oblique veins while 5e shows 4. If so, this will affect the diagnosis in line 235.

LIne 269: specify left lateral view as in Fig. 6 caption.

Line 273: How was genitalia similarity based and on which characters? To my eyes, A. pantescus and A. confalonierii have similar ectoparameres but different lateral lobes of the epiphallus. The lateral lobes of the epiphallus rather resemble those of A. turcomanoides.

Lines 323-324: these temporal measurements depend on temperature. Note if temperature is known or not.

Line 329: This description is confusing. Do the authors mean to call attention to the first two to four syllables in a crescendo? Identifying a crescendo in one of the song figures would help.

Line 332: Impact vs. pulse. vs. syllable must be defined in the Methods, with reference to acoustical terminology. If a pulse is a toothstrike then pulse may be synonymous with impact. That would make the entire unit in Fig. 10c a syllable, consistent terminology if that unit is produced by one cycle of wing movement. That would change pulses as defined here to hemi-syllables (Baker & Chesmore 2020). Other authors consider a pulse to be the entire unit in Fig. 10c (e.g. Weissman & Gray 2019).

Fig. 10c does not directly count impacts. Was this done manually?

Line 344: cite Fig. 10d. Cite throughout when describing acoustical characters and measurements, e.g. also line 350 cite Fig. 10e.

Line 353: Results not shown, please cite as such.

Line 357: technogenic effects of spectrum measurement were mentioned in line 349. The July measurement was made with a device with wider spectral coverage. A higher measured fundamental frequency could also be a technogenic effect, if recordings with lower frequency limits biased frequencies downward. With the small sample size of recordings this could also be due to individual variation.

Table 3 Show sample sizes for each descriptive statistic. This is presumably the subset of recordings in Table 2.

Discussion

Line 458: Not sure what this sentence is intended to communicate. Does this refer to the crickets or the plants?

Nice species origin hypotheses.

The Discussion could circle back to the value of passive acoustical recording for species discovery far from the taxa for which monitoring was intended in the first place.

Island cricket faunas have received much attention (e.g. Shaw lab with Hawaiian crickets, Otte & Perez-Gelabert with Caribbean crickets), the cricket fauna of this island may be placed in to broader island biogeographical context by citing such work.

Round 2

Reviewer 1 Report

The authors have greatly improved the manuscript, for the text and illustrations, for acoustic analyses and taxonomic description.

They could have sequenced at least one specimen of the new species and propose the sequence as a set of characters, even without making a phylogeny of the whole genus.

I recommand publication as such.